# Investigating the dynamics of climate finance disbursements: A panel data approach from 2003 to 2022

**Mohamed Ibrahim Nor**[ID]*

Institute of Climate and Environment –ICE, SIMAD UNIVERSITY, Mogadishu, Somalia

* m.ibrahim@simad.edu.so

## Abstract

This study investigates the intricate dynamics of international multilateral climate finance disbursements from 2003 to 2022 via an extensive dataset from the Climate Funds Update (CFU). By employing panel data econometric models, including pooled ordinary least squares (OLS), fixed effects (FE), and random effects (RE) models, the study elucidates the impact of grants and approved funds on disbursement levels across different income groups. The analysis reveals that while grants do not significantly influence disbursements, the approval of funds plays a critical role in enhancing disbursement efficiency. The random effects model, validated through the Hausman test, emerges as the optimal model for this context. The findings underscore the importance of streamlined approval processes in ensuring effective climate finance disbursements and highlight the need for further investigation into the non-significance of grants. The forecasting results indicate a positive trend in disbursements from 2023 to 2027, with potential fluctuations driven by external factors. This study provides valuable insights for policymakers and stakeholders to optimize climate finance mechanisms and improve fund utilization for sustainable development.

## Introduction

Climate change is one of the most critical issues that the world faces today, with wide-ranging and far-reaching effects on all forms of life on the mother planet [1]. The rising global temperatures, increased incidence of extreme weather events, sea-level rise and threats to food security highlight the need for a comprehensive approach to address this problem [2]. There is considerable scientific evidence that shows that temperature patterns have changed over time due to human activities such as burning fossil fuels and emitting heat-trapping gases such as carbon dioxide [3,4]. Although climate change affects every corner of the globe, developing countries are more vulnerable than other countries are because of their limited adaptive capacity [5]. These weak communities often lack the resources or infrastructure necessary for them to adapt themselves to these disasters, which include droughts and floods, among other weather extremes caused by climate change [5]. Moreover, these regions are further burdened by various socioeconomic challenges, making them susceptible to the negative impacts of climate change, thus hampering global progress toward the SDGs [5].

**Data availability statement:** All data utilized in this study are included in the manuscript and/or its Supporting information files. The data were sourced from publicly available datasets provided by Climate Funds Update, a respected Washington-based think tank that tracks and analyzes climate finance. These data are accessible via their website: https://climate-fundsupdate.org.

**Funding:** Center for Research and Development (CRD), SIMAD University.

**Competing interests:** The authors have stated that they have no conflicts of interest to disclose.

Climate change problems are multifaceted rather than just environmental but also socio-political in nature, particularly in underdeveloped areas [6,7]. The persistent increase in global temperatures puts livelihoods at risk by increasing food insecurity and threatening security, especially in places already experiencing political instability and economic vulnerabilities [8–10]. These areas face greater risk of climate-induced migration, as people abandon their homes due to extreme weather conditions, thus exerting pressure on host communities and potentially leading to conflict situations [11–13]. Climate change affects developing nations more than any other group of countries does, indicating a global inequality in the ability to adapt and mitigate its impacts [5]. These regions often lack financial resources, infrastructure and technology and are unable to respond effectively to climate change challenges, hence becoming more vulnerable and resulting in negative socioeconomic consequences [14]. This disparity in adaptive capacity requires immediate attention if global climate goals are to be realized fairly and sustainably.

International climate finance has become an important means for addressing this gap by providing the financial resources necessary for addressing climate change in developing countries [15]. The effectiveness of these financial mechanisms must be considered very carefully if limited funds are to be used wisely and evenly to achieve anticipated outcomes. Given the complexities associated with the allocation and use of climate financing, they should be subjected to stringent assessment and evaluation, thus maximizing their contributions to global climate goals [16,17].

The number of studies that critically analyze whether these financial mechanisms are effective in developing countries remains relatively small despite a growing body of research on climate finance. Most existing studies investigate broad trends and patterns regarding climate finance but often ignore particular challenges as well as needs for communities from least developed countries. For example, targeting the unique circumstances plus priorities compared with other regions signifies a gap in the literature for this case. Moreover, research on the effects of various financial instruments and mechanisms on the effectiveness of climate finance is limited. Although grants, loans and public–private partnerships have been studied for their advantages and disadvantages, a comprehensive review that compares them with those of developing countries is lacking [18,19]. The existing gap in knowledge highlights the need for more detailed and context-specific studies that can be used to inform more effective climate finance strategies. The objective of this study is to investigate how climate finance mechanisms are effective and critically assess how to improve the current approach toward climate financing, hence adding value to a wider body of knowledge concerning climate financing. By adopting an interdisciplinary approach drawing from ideas across multiple domains, this research will facilitate a deeper understanding of the intricacies involved in addressing issues related to climatic finances. Hence, this study investigates the allocation patterns of global climate funds from 2003--2022, focusing on grants, approved funds, and disbursed funds.

The importance of this study is demonstrated by the urgency of addressing how vulnerable communities in developing countries are disproportionately impacted by climate change. Even with international climate finance available, there is an urgent need to ensure that these resources can be used efficiently to increase the resilience of these communities and mitigate the effects of climate [14]. This research identifies good practices as well as areas requiring improvement to increase the effectiveness of mechanisms for channeling money into fighting climatic variability, hence making them responsive to the needs and priorities of vulnerable people [16,17]. Therefore, sustainable development goals necessitate this move while establishing global climate resilience.

The increasingly frequent disaster caused by the negative impacts of global warming on fragile populations, particularly those from underdeveloped economies, demonstrates that

action regarding global environmental protection should be taken urgently [7,20]. The theoretical framework for this study is grounded in the conceptual frameworks of environmental justice and sustainable development, which advocate for equitable and inclusive approaches in climate governance [5]. In this context, this study therefore examines how international finance for environmental conservation leads to the creation of appropriate policies that promote fairness in addition to sustainability.

It is also important because it will contribute to informing policy makers at both the national and international levels. This research helps identify gaps in existing climate finance mechanisms through critical analysis so that ways can be found where there are no suggestions concerning the allocation and use of financial resources [14]. This guarantees that climate finance effectively supports the adaptation and mitigation efforts of vulnerable communities and hence increases their resilience to climate change [16,17]. Therefore, this study is significant in terms of informing effective and equitable climate finance strategies.

In summary, this study investigates the allocation patterns of global climate funds from 2003–2022, focusing on grants, approved funds, and disbursed funds. The urgent need to evaluate and optimize the allocation and utilization of international climate finance is paramount in supporting developing countries in their fight against climate change. This research identifies significant gaps in the literature regarding the effectiveness of different financial mechanisms and their impact on vulnerable communities. By filling these gaps, this study aims to contribute valuable insights and recommendations that can inform more equitable and effective climate finance strategies. This, in turn, will enhance the resilience of developing countries to climate change, supporting global efforts to achieve sustainable development goals and fostering climate resilience.

## Literature review

### Evolving trends in climate finance

The evolution of climate finance has been a central theme across recent United Nations Climate Change Conferences, reflecting a growing recognition of the financial imperatives associated with climate action. At the Paris Agreement of 2015, countries, developed ones, in particular, made a pledge concerning a transfer of 100 billion dollars, yearly, starting from the year 2020, in order to help developing countries with avoidance and adaptation of climate change, but even this pledge that was made seemed to be insufficient at the subsequent conferences that concerned the worsening circumstances associated with climate change. COP26 in Glasgow (2021) is a good example, but it was made clear that something will have to change in the way financial resources are supplied. During the following year a lot changed, beginning at COP27 that was held at the beginning of the conference in Sharm El-Sheikh (2022) the Loss and Damage Fund, a fund that supplements existing insurance mechanisms for some vulnerable countries that would suffer damage as a consequence of climate change, was created. This change was significant because it was the first time that the global problems associated climate change were framed in terms of disempowerment.

COP28 in Dubai (2023) and COP29 in Baku (2024) developed these trends by increasing attention on mobilization of climate finance. COP28 reached a major deal on how to move away from fossil fuels and how to increase renewable energy capacity by three times and pointed out the financial obligations required for that. At COP29, the developed countries agreed to provide on average $300 billion each year by 2035 to help developing countries deal with climatic issues". While the figure exceeds earlier promises, this figure is still below the 1.3 trillion dollars claimed by the developing nations and thus the friction and complexities associated with mobilising adequate resources

remain. These conferences have also focused on the implementation of carbon markets and the Loss and Damage Fund aimed at increasing climate finance accountability and efficiency. Nevertheless, the still existing gap between available and needed funds underlines the urgent requirement of integrating novel funding mechanisms and greater collaboration on international scale to fulfil the increasing requirements of climate action worldwide.

## Impacts and risks of climate change and climate finance

The issue of climate change presents an urgent and significant global challenge, supported by an increasing body of scientific evidence that highlights its extensive and adverse impacts [21]. The ongoing increase in global temperatures is accompanied by a notable increase in the occurrence of severe weather phenomena, including hurricanes, wildfires, and droughts, which inflict significant damage to both human settlements and physical infrastructure [22]. The continuous progression of sea-level rise poses a significant threat to coastal communities and intensifies the occurrence of flood events [23]. Ecosystems are currently experiencing significant challenges, as numerous species are confronted with the threat of extinction, thereby jeopardizing the stability of food security [24].

The consequences of these impacts extend beyond national boundaries and have significant economic and social implications for both developed and developing countries [25]. This is because uncontrolled climate change has led to an increase in the frequency and intensity of extreme weather events, adverse health effects, and disruptions to the economy [26]. The utilization of empirical evaluations to determine vulnerability to climate change has been instrumental in guiding the allocation of climate finance resources to regions and communities that require them the most [27]. By identifying areas of high vulnerability, climate finance can be allocated more effectively, ensuring that limited resources are channeled to those who require them the most [28]. These processes produce empirical evidence that illuminates the tangible impact and effectiveness of climate finance initiatives in attaining their desired objectives, such as reducing emissions, adapting to climate change, or promoting sustainable development [29]. These requirements encompass a wide range of approaches, ranging from enhancing the ability of vital infrastructure to withstand challenges to ensuring the security of food supplies in response to changing weather conditions [30].

Coastal regions that are particularly susceptible to the impacts of rising sea levels and intensified extreme weather events should prioritize preparedness measures to safeguard both human lives and economic activities [31]. The aforementioned targets, which are based on empirical research, play a crucial role in the worldwide effort to combat climate change [32]. These assessments offer a comprehensive analysis of the distinct difficulties and vulnerabilities experienced by various geographical areas and demographic groups, considering variables such as susceptibility to climate-related threats, socioeconomic circumstances, and the ability to adapt [28]. These assessments offer a somber depiction of the repercussions of unregulated emissions and contribute to the ambitious objectives outlined in global agreements. They provide essential benchmarks that direct nations toward coordinated efforts, stimulate innovation, and foster collaboration. Ultimately, their aim is to ensure a future that is both sustainable and resilient to climate-related challenges for all individuals [33].

The multifaceted impacts of climate change, ranging from extreme weather events to threats to ecosystems and food security, underscore the urgent need for targeted climate finance. By assessing vulnerabilities and allocating resources to the most affected regions and communities, climate finance plays a pivotal role in mitigating these risks and building

resilience. The evidence reveals that unchecked climate change will exacerbate socioeconomic disparities and cross-border challenges, making a coordinated global response indispensable. Effective use of financial mechanisms ensures that climate adaptation and mitigation strategies are both equitable and impactful, addressing immediate threats while fostering long-term sustainability.

## Economic analyses and climate finance

The empirical research conducted thus far has provided valuable insights into the financial implications of mitigating climate change through the reduction of greenhouse gas emissions [34]. Although the initial capital investments necessary for such undertakings are significant, they are indispensable investments in the future characterized by sustainability and resilience [35]. Over the course of time, the expenses incurred from not taking action surpass the costs associated with proactive mitigation measures [36]. Economic models and analyses have highlighted the considerable potential economic advantages associated with implementing proactive strategies to mitigate climate change [37].

The transition toward a low-carbon economy not only strongly aligns with the principles of environmental sustainability but also presents a wide array of benefits [38]. Cost-benefit analyses are indispensable instruments for assessing the economic feasibility of climate finance initiatives [39]. Cost-benefit analyses offer decision-makers a comprehensive understanding of the economic justification for funding climate mitigation and adaptation endeavors by quantifying both the initial investments and the potential future savings linked to diminished climate impacts [40].

The reduction in greenhouse gas emissions has the potential to generate significant cost savings in healthcare through the mitigation of illnesses associated with air pollution and the enhancement of public health [41]. This transition has the potential to facilitate the generation of employment opportunities by means of investments in renewable energy, energy efficiency, and environmentally friendly technologies, thereby promoting economic expansion while simultaneously addressing the consequences of climate change [42]. The aforementioned findings highlight the persuasive economic rationale for taking action on climate change, illustrating the compatibility of sustainability and economic well-being [43]. The aforementioned analyses thoroughly evaluate the financial obligations necessary to sustain climate action in comparison to the potential long-term benefits derived from mitigating climate-related damage [44].

The utilization of data-driven methods not only serves to augment transparency and accountability in the realm of climate finance but also guarantees that scarce resources are allocated to endeavors that possess the highest capacity to significantly mitigate the pressing issues presented by climate change [45]. These initiatives assist in the allocation of financial resources toward the goal of mitigating climate change. They highlight the ethical obligation as well as the prudent economic rationale for investing in a future that is both sustainable and resilient [46].

Economic analyses provide compelling evidence that the costs of inaction on climate change far exceed the investments required for proactive mitigation and adaptation. The transition to a low-carbon economy offers significant financial, social, and health benefits, reinforcing the alignment between sustainability and economic prosperity. Cost-benefit analyses further validate the fiscal prudence of investing in renewable energy, energy efficiency, and resilient infrastructure, highlighting the potential for significant long-term savings and economic growth. Climate finance emerges as a critical tool in transforming economies by not only addressing environmental challenges but also unlocking opportunities for innovation and development.

## Mitigation, adaptation, policy response, and climate finance

It is crucial to prioritize immediate and collaborative actions aimed at mitigating and adapting to climate change to protect the welfare of both present and future generations [47]. Existing literature emphasizes transformative measures such as transitioning from fossil fuels to renewable energy sources, enhancing energy efficiency across various sectors, and implementing integrated strategies to achieve substantial emission reductions [48]. Several studies have played crucial roles in identifying the urgent adaptation requirements of vulnerable communities and nations that are struggling with the persistent effects of climate change [49]. Acknowledging the significance of both adaptation and mitigation, these studies emphasize the necessity of international collaboration and assistance to ensure that individuals who are most susceptible are adequately prepared to confront the difficulties posed by a shifting climate [50].

The utilization of empirical research has been instrumental in influencing the establishment of global emission reduction targets, as demonstrated by the significant Paris Agreement [51]. The Paris Agreement emphasizes the necessity of undertaking endeavors to restrict global warming to 1.5 degrees Celsius, as there are notable disparities in the consequences between a temperature increase of 1.5°C and 2°C [52]. Empirical evidence derived from case studies of climate finance initiatives and projects demonstrates the transformative efficacy of such funding [53]. These success stories effectively demonstrate the transformative potential of international multilateral climate finance. They showcase how such financial support can serve as a driving force for constructive transformations, empowering nations to not only mitigate greenhouse gas emissions but also effectively address the impacts of climate change [54]. By presenting these accomplishments, we not only commemorate notable attainments but also foster a sense of motivation to sustain financial backing and assistance for climate endeavors that offer the potential for a more environmentally sustainable and resilient future for all [55].

Numerous public opinion polls and surveys have consistently indicated a pervasive endorsement for global initiatives aimed at addressing the issue of climate change [56]. Therefore, these empirical evaluations highlight how crucial it is to allocate funds to reduce greenhouse gas emissions and establish a low-carbon, environmentally conscious global economy [57]. This will promote the development of resilience and sustainable practices at a worldwide level [58]. In addition, Furthermore, the process of diversifying energy sources serves to enhance energy security by diminishing susceptibility to fluctuations in fossil fuel prices and disruptions in supply. The aforementioned targets are based on meticulous scientific evaluations that highlight the pressing necessity of constraining global warming to a level significantly lower than 2 degrees Celsius above the temperatures recorded during the preindustrial era [59].

Moreover, these efforts align with their broader sustainable development objectives. The aforementioned examples highlight the significant and concrete advantages that can be achieved through targeted climate financing, such as renewable energy initiatives that provide clean electricity to isolated communities, reforestation endeavors that increase carbon sequestration and biodiversity, and measures aimed at enhancing the resilience of vulnerable populations against climate-related impacts [55]. The implementation of this focused strategy not only strengthens the ability of susceptible communities to withstand adverse conditions but also optimizes the beneficial outcomes of financial resources allocated toward climate-related initiatives, thereby promoting a more just and enduring approach to addressing the worldwide climate emergency [60].

The continuous monitoring and evaluation of climate finance projects and programs are integral elements in the implementation of an efficient response to the climate crisis [29,53]

The presence of empirical evidence regarding public concern has a substantial influence on shaping political decisions and commitments pertaining to climate finance. With respect to the intensifying climate crisis voiced by individuals across the globe, there is growing pressure on political leaders to address these concerns by implementing comprehensive climate policies and making substantial financial investments [61]. The alignment of public sentiment with the necessity of taking action on climate change generates a potent synergy that has the potential to result in increased and enduring financial support for climate finance endeavors [62]. Through a comprehensive evaluation of project outcomes and diligent monitoring of progress, individuals in positions of authority can discern effective and ineffective approaches, thereby facilitating the optimization of strategies and the efficient allocation of resources. This underscores the pivotal importance of public awareness and involvement in worldwide endeavors to address this urgent issue.

Global efforts to combat climate change through mitigation, adaptation, and robust policy frameworks are significantly bolstered by targeted climate finance. From achieving the ambitious goals set by the Paris Agreement to fostering international collaboration and innovation, financial mechanisms enable countries to implement transformative strategies. By addressing both immediate adaptation needs and long-term mitigation goals, climate finance ensures that no community is left behind in the fight against climate change. Ultimately, the integration of financial, policy, and scientific efforts underscores a collective commitment to safeguarding a sustainable and resilient future for all.

## Theoretical foundation for climate finance

The study of international multilateral climate finance is enriched by applying various theoretical frameworks that provide deeper insights into the complexities of funding mechanisms. **Institutional theory** illuminates the governance roles of international institutions in climate finance, **resource mobilization theory** identifies key factors influencing funding availability, and **environmental justice frameworks** emphasize the critical need for equitable distribution of climate finance resources.

1. Institutional theory.  Institutional theory provides a robust framework for understanding the influence of formal and informal institutions in the governance of international multilateral climate finance [63–65]. This theory underscores the critical role of institutions, such as the United Nations Framework Convention on Climate Change (UNFCCC) and the Green Climate Fund (GCF), in shaping the allocation and distribution of climate finance resources. These institutions establish norms, regulations, and protocols that govern financial flows, impacting decision-making and policy implementation processes [66]. Their governance mechanisms are pivotal for ensuring effective and equitable resource allocation that supports sustainable development and enhances climate resilience. As emphasized by Scott (2008), institutional design significantly affects transparency, accountability, and legitimacy, which are essential for fostering trust among stakeholders, including donors, recipients, and civil society.

Furthermore, institutional theory highlights the importance of collaboration and inclusivity in climate finance governance. Institutions like the UNFCCC and GCF provide platforms for negotiation and consensus-building among nations, promoting cooperation and mobilizing financial resources for climate action [67]. Transparent governance frameworks and accountability mechanisms not only ensure proper fund utilization but also enhance confidence in the fair distribution of resources, particularly for marginalized and vulnerable communities [68]. By incorporating diverse stakeholder interests, these institutions enhance the legitimacy of climate finance processes, aligning them with broader social and environmental goals (Betsill & Bulkeley, 2004). Consequently, institutional theory sheds light on how

international organizations can effectively address the complexities of climate finance governance while fostering inclusivity and equity.

2. Resource mobilization theory.  Resource mobilization theory provides a valuable framework for analysing the processes through which financial resources are gathered and distributed for climate finance initiatives. This theory examines the various factors influencing the mobilization and allocation of funds for climate change mitigation and adaptation projects, highlighting the critical role of stakeholders such as donor nations, international institutions, and private sector entities [69,70]. It also underscores the importance of addressing discrepancies in funding availability and identifying the challenges inherent in mobilizing adequate resources for climate projects [18,62]. A key insight from resource mobilization theory is the necessity of addressing barriers such as the lack of suitable financial instruments and market inefficiencies, which limit private sector participation in climate-related initiatives [71,72]. Policymakers can mitigate these challenges by implementing regulatory reforms, fostering capacity-building programs, and developing tailored financial tools that cater to the specific needs of climate projects [73,74].

Innovative financial mechanisms also play a pivotal role in expanding the accessibility and efficiency of climate finance. Instruments such as climate bonds and public-private partnerships have emerged as effective strategies for leveraging private sector investment in climate initiatives, thus addressing the resource gap [73,74]. These mechanisms not only broaden the scope of resource mobilization but also enhance stakeholder engagement, drawing in private investors, philanthropic organizations, and government entities (Ballesteros, Nakhooda et al., 2010; Karanth & Archer, 2018). According to [75–77], the adoption of such inventive methodologies is essential for bridging the deficit in climate finance, enabling transformative change and fostering resilience to climate challenges. By facilitating the alignment of diverse financial sources, resource mobilization theory provides critical insights for advancing global climate objectives and promoting sustainable development [78,79].

3. Environmental justice frameworks.  Environmental justice frameworks provide a critical lens for evaluating the fairness and equity of climate finance, particularly in addressing the needs of marginalized and vulnerable communities. These frameworks emphasize principles such as equity and just transitions, ensuring that the benefits of climate action are distributed in a manner that mitigates existing social and environmental disparities [80,81]. Within the context of climate change, marginalized communities often face disproportionate negative impacts while lacking access to the resources necessary for adaptation and mitigation [82,83]. To address these inequities, environmental justice frameworks advocate for the removal of institutional and geographic barriers that hinder access to climate finance, ensuring fair distribution to prevent vulnerable groups from being left behind [84]. This ethical obligation is essential for fostering inclusivity and promoting equitable outcomes in global climate finance initiatives [85].

The integration of equity, inclusivity, and community engagement into climate finance mechanisms is a central tenet of environmental justice. Equitable allocation of resources must consider historical accountability for greenhouse gas emissions, providing targeted support to those most affected by climate change [86]. Furthermore, inclusivity requires deliberate efforts to involve marginalized communities in decision-making processes, amplifying their voices and ensuring that their needs are addressed [87]. Genuine community engagement goes beyond superficial involvement, prioritizing meaningful participation in the planning and implementation of climate initiatives [88]. By addressing these aspects, environmental justice frameworks not only facilitate fair and sustainable progress but also enhance the legitimacy and effectiveness of climate finance mechanisms, contributing to a more resilient and low-carbon future [85,89].

### Research gaps and study rationale

Despite the growing interest in climate finance, a significant gap remains in critically examining the effectiveness of financial mechanisms in addressing the priorities and needs of communities in least developed countries. The existing literature largely emphasises on broad trends and patterns, often ignoring the unique challenges faced by least developed countries (LDCs). For instance, while grants, loans, and public–private partnerships have been studied for their generic advantages and disadvantages, their contextual applicability and effectiveness in these regions have not been comprehensively compared. This gap is particularly striking given the heightened vulnerability of these communities to climate change and the inadequacy of current financial mechanisms to address their urgent needs. Furthermore, there is a shortage of literature examining allocation patterns of global climate funds—particularly the dynamics of approved versus disbursed funds over time—and how these patterns align with the resilience-building objectives of developing nations. This lack of context-specific analysis underscores the need for more nuanced research to ensure that financial mechanisms are tailored to effectively meet the demands of communities disproportionately affected by climate change.

This study addresses critical research gaps by investigating the allocation patterns of global climate funds from 2003 to 2022, with a particular focus on grants, approved funds, and disbursed funds. By adopting an interdisciplinary approach that integrates perspectives from development economics, environmental policy, and climate resilience, this research provides a deeper understanding of the effectiveness of various financial instruments in supporting vulnerable communities. The urgency of this endeavour is underscored by the disproportionate impacts of climate change on LDCs, where international climate finance plays a pivotal role in enhancing resilience and mitigating adverse effects. Identifying good practices and pinpointing areas for improvement will contribute to the development of more responsive and impactful climate finance mechanisms, aligning with the Sustainable Development Goals and global climate resilience initiatives. Through its focus on the intersection of finance, climate policy, and development, this study aims to inform more equitable and efficient strategies for channelling resources into regions where they are most needed.

## Research methodology

### Type and source of data

The study utilizes an extensive dataset from the **Climate Funds Update (CFU)**[1] spanning the period from 2003--2022, which encompasses detailed records of international multilateral climate finance. This dataset provides comprehensive information on grants, approved funds, and disbursed funds, enabling a thorough analysis of financial flows in the context of global climate initiatives. By leveraging these longitudinal data, this study aims to elucidate the dynamics and efficacy of climate finance allocation, examining how different funding mechanisms impact the disbursement of resources. The inclusion of a two-decade timespan allows for the observation of trends and patterns over time, offering valuable insights into the evolving landscape of climate finance and its implications for sustainable development. The data are classified into various income groups as defined by the World Bank, namely:

a) High-income countries (HICs).

b) Upper Middle-Income Countries (UMICs).

c) Lower Middle-Income Countries (LMICs).

d) Low-Income Countries (LICs)

e) Unspecified Income Group

**Climate Funds Update**, a respectable Washington-based think tank that tracks and analyzes climate finance. The data was obtained by means of their publicly accessible website, namely https://climatefundsupdate.org

**Variables:**

a) Grants: Total amount of grants approved for climate finance projects.

b) Approved Funds: Total amount of funds approved for disbursement.

c) Disbursed Funds: Total amount of funds actually disbursed.

The dataset is structured as a panel dataset, where each observation corresponds to a specific income classification and year combination, allowing for the examination of the dynamics over time across different income groups.

## Model specifications

To analyze the impact of grants and approved funds on the levels of disbursement, we use three different panel data models: pooled ordinary least squares (OLS), fixed effects (FE), and random effects (RE) models. The general form of the model is specified as follows:

$$\text{Disbursed Funds}_{jt} = \beta_0 + \beta_1 \text{Grants}_{jt} + \beta_2 \text{Approvals}_{jt} + \gamma x_{jt} + u_j + \varepsilon_{jt}$$

where:

- $j$ denotes the income classification group

- $t$ denotes the year

- Disbursed Funds$_{jt}$ is the dependent variable representing the total funds disbursed

- Grants$_{jt}$ is the independent variable representing the total grants

- Approved Funds$_{jt}$ is the independent variable representing the total number of approved funds

- $X_{jt}$ is a vector of control variables, which may include other relevant factors affecting disbursement levels.

- $u_j$ is the income group-specific effect

- $\varepsilon_{jt}$ is the idiosyncratic error term

## Estimation procedure

Pooled OLS model. The pooled ordinary least squares (OLS) model operates under the assumption that there are no unique or inherent differences among income groups within the measurement set, thereby treating all observations as part of a single, homogeneous dataset. By aggregating the data across different income groups and ignoring potential group-specific effects, this approach allows for straightforward application of OLS regression to the combined dataset. Consequently, the pooled OLS model provides a general overview of the relationships between the dependent and independent variables without accounting for the possibility that variations within income groups might influence the overall outcomes. While this method simplifies the analysis and can offer broad insights, it may overlook critical nuances and heterogeneities specific to individual income groups, potentially leading to biased

or incomplete interpretations of the underlying data dynamics. This model pools all the data and runs a simple OLS regression on the combined dataset:

$$\text{Disbursed Funds}_{jt} = \beta_0 + \beta_1 \text{Grants}_{jt} + \beta_2 \text{Approvals}_{jt} + \varepsilon_{jt}$$

Fixed effects model. The fixed effects model addresses potential biases by accounting for income group-specific characteristics that may influence the dependent variable. This approach controls for time-invariant attributes within each income group by incorporating unique intercepts for each group, thus isolating the impact of variables that vary over time. By allowing each income group to have its own intercept, the fixed effects model effectively controls for unobserved heterogeneity, ensuring that the estimated relationships between the independent variables and the dependent variable are not confounded by these constant characteristics. Consequently, this model provides a more nuanced and accurate depiction of the effects of the independent variables on the dependent variable, as it isolates the impact of within-group variations over time from the potentially confounding influences of between-group differences. It controls for time-invariant characteristics by allowing a unique intercept for each income group:

$$\text{Disbused Funds}_{jt} = \beta_0 + \beta_1 \text{Grants}_{jt} + \beta_2 \text{Approvals}_{jt} + u_j + \varepsilon_{jt}$$

where $u_j$ represents the income group-specific effects, capturing all time-invariant differences between income groups.

Random effects model. The random effects model operates under the assumption that the income group-specific effects are uncorrelated with the independent variables included in the regression. This approach allows for the inclusion of both within-group and between-group variations by incorporating a random intercept that captures the unobserved heterogeneity across different income groups. Unlike the fixed effects model, which controls for all time-invariant differences by allowing each group to have its own intercept, the random effects model assumes that these group-specific effects are randomly distributed and not systematically related to the independent variables. This assumption enables the model to utilize the entire dataset more efficiently, providing a balance between the fixed effects model's control for unobserved heterogeneity and the pooled OLS model's simplicity, potentially leading to more efficient estimates if the assumption holds true. This model includes a composite error term:

$$\text{Disbursed Funds}_{jt} = \beta_0 + \beta_1 \text{Grants}_{jt} + \beta_2 \text{Approvals}_{jt} + u_j + \varepsilon_{jt}$$

In this model, $u_j$ is assumed to be randomly distributed across the income groups.

## Empirical findings

### Trend analysis

Fund allocation and focus. From 2003 to 2022, approximately 3,067 climate projects were approved by multilateral climate finance sources and funded, highlighting the global commitment to mitigate climate change through various initiatives. These are sector-specific and cross-sectoral projects that focus on mitigation and adaptation approaches, resulting in significant contributions toward reducing greenhouse gases, enhancing resilience and promoting sustainable development. Allocating funds show that approximately 60.45% of the funds are multifaceted in nature, as they employ multiple financial instruments and strategies

to address the complex challenges of climate change comprehensively. Moreover, general mitigation efforts constitute 16.15%, whereas REDD-specific mitigation accounts for 11.72% of the funding along with adaptation initiatives at 11.67%, thus exemplifying a combination of proactive and reactive measures within climate finance (see Fig 1).

Disbursement disparities: The Climate funding gap. The review of international multilateral climate financing between 2003 and 2022 reveals that there is an enormous gap between the amount pledged to be given out as funds for this purpose and those that were disbursed, making it difficult to allocate financial resources efficiently for climate matters. This means that only $28,380 million out of $43,183.86 million were approved for spending, translating into a variance of approximately 34%. Similarly, disbursement figures are even more depressing because only $10,187.73 was actually disbursed, meaning that there was an enormous financial deficit amounting to approximately 76.41% (see Fig 2). This discrepancy highlights systemic inefficiencies as well as bottlenecks in approving or disbursing processes by which money for addressing climatic impacts reaches developing countries, especially vulnerable ones, at the right time or effectively implemented over time. This shortfall should be closed so that pledged monies translate into concrete actions against climatic issues, improving global resilience as well as fostering sustainability drives worldwide, while these obstacles can be removed through further efficient, transparent and streamlined mechanisms that will hasten approval and disbursement processes for fully actualizing financial commitments in combating the urgent impacts of climate change.

Funding gaps by country/region. The evaluation of international multilateral climate finance from 2003--2022 clearly shows major discrepancies in disbursed fund allocations, thus leaving the highest proportion of vulnerable countries at a disadvantage. Least developed countries (LDCs) received only $2,610.33 million out of a total disbursed amounting to $10,658.24 million, representing a 24% share, indicating the existence of crucial unmet

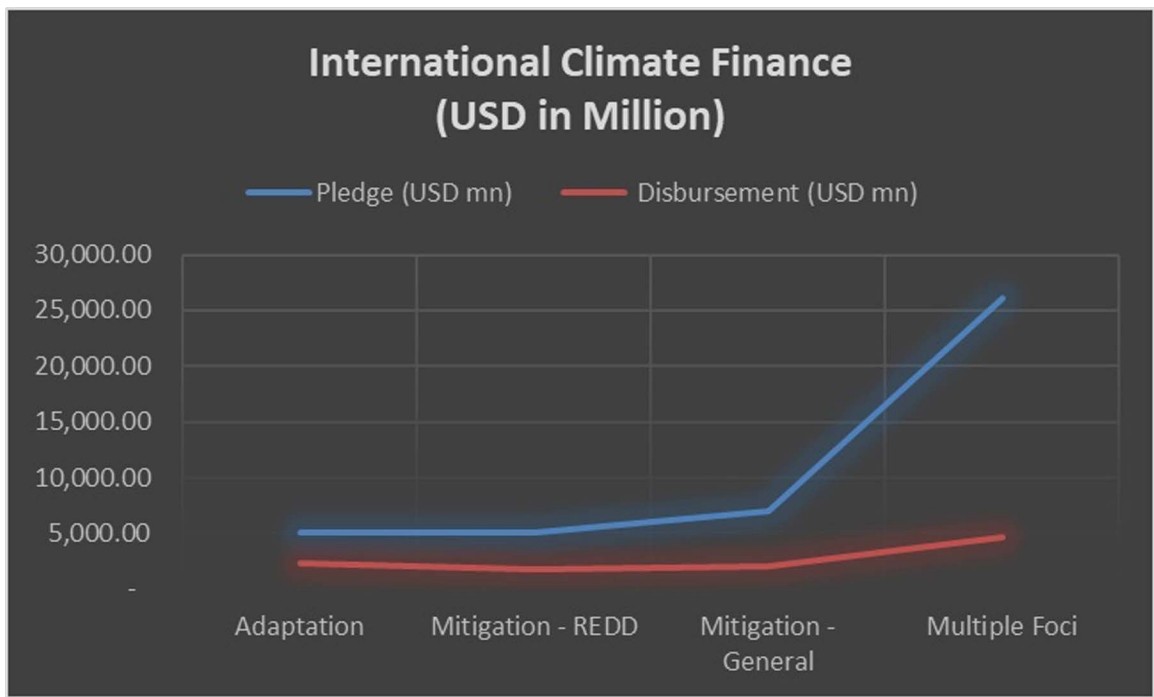

**Fig 1.  International Climate Finance (USD in Million).**

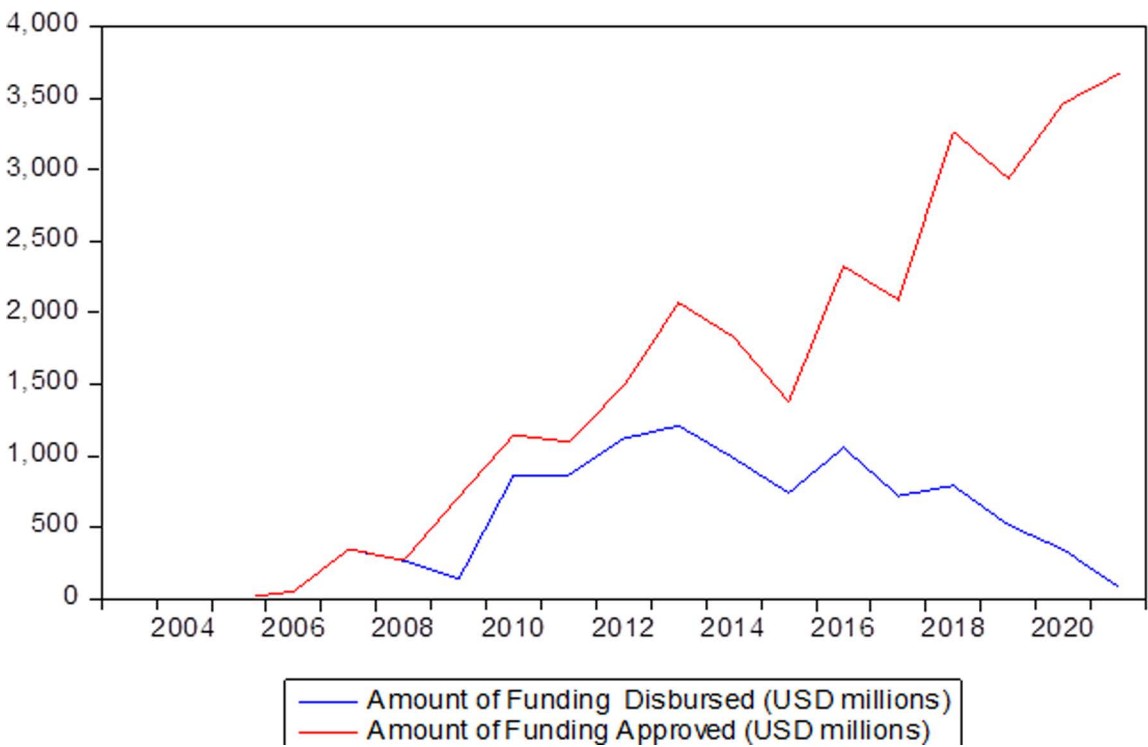

**Fig 2. Amount of Funding Disbursed/Approved.**

funding needs and imbalanced distribution (see Fig 3). Similarly, Small Island Developing States (SIDs) as well as Fragile or Conflict-Affected States have just given away 10% and 12%, respectively, of all the disbursements made thus far. These statistics demonstrate how important it is to allocate more resources on an equitable basis for regions experiencing heightened climate vulnerabilities, including rising sea levels, extreme weather events, and compound challenges arising from instability and conflict. It is imperative to address these funding disparities to improve inclusive national systems that can cope with climatic changes in such nations, which are sufficiently supported in terms of mitigating both the environmental effects linked to climatic forces and attaining sustainable growth paths.

Climate finance efficacy. The efficiency ratio, a key yardstick of climate finance effectiveness, differs greatly between pledged funds and disbursed funds within international multilateral climate finance from 2003--2022. These data show that there are significant weaknesses in the allocation and utilization of resources for environmental protection. Among other issues, this ratio implies that only one out of every four dollars promised was handed over, highlighting an important deficit in financial translation into actions against climate change. This low disbursement rate calls for improved mechanisms to increase the efficiency, transparency and efficacy of climate finance processes (see Fig 4). To ensure that committed capital is used most effectively to meet climate targets while assisting vulnerable nations in mitigating and adapting to climatic conditions, constant review and analysis of efficiency ratios should be undertaken.

Climate financing by income classification. This dataset presents full information on grants approved and funded, allowing for comprehensive financial flow analysis in the context of global climate initiatives. Climate finance trends have indicated that high-income countries

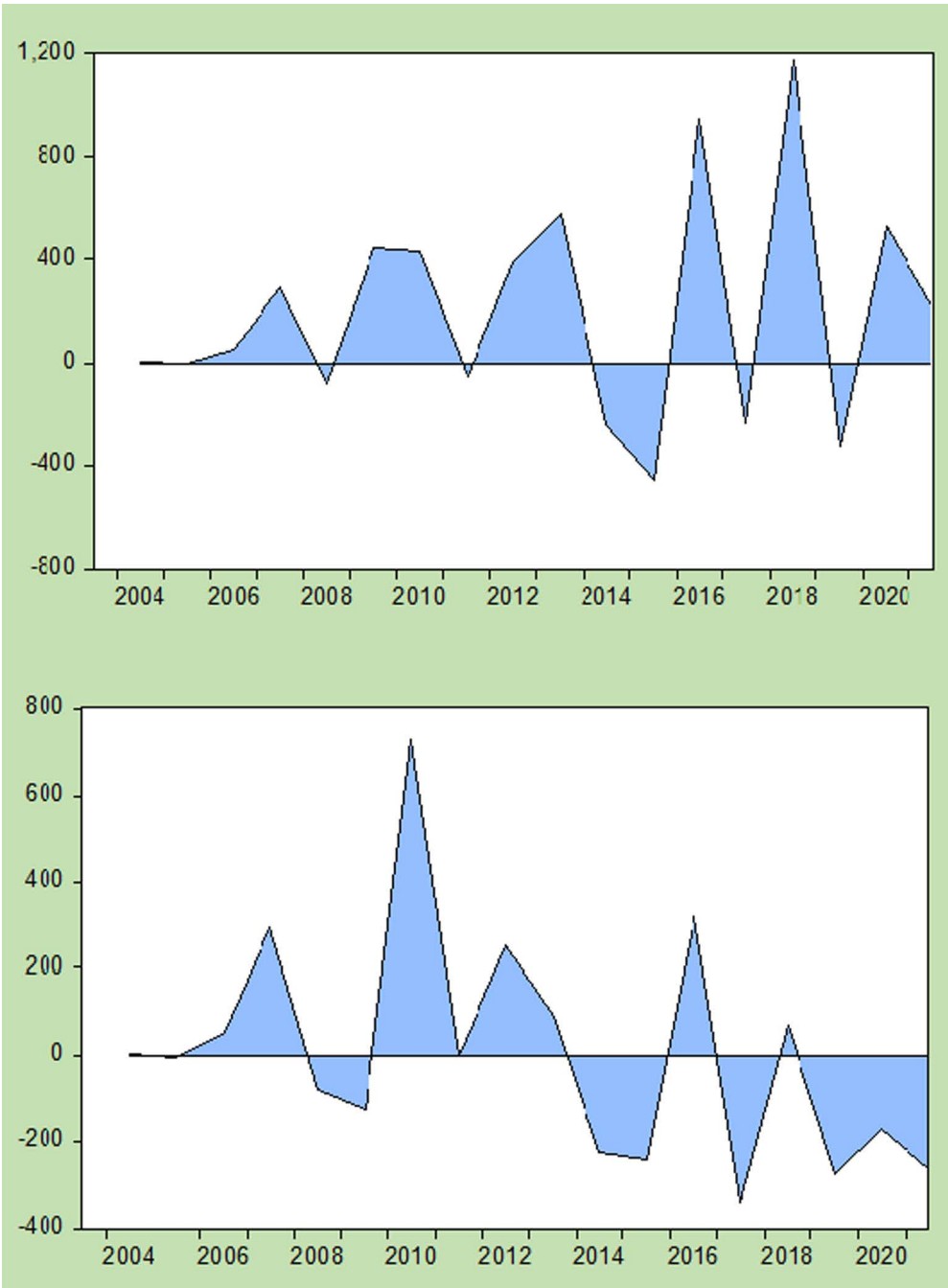

**Fig 3. Differenced Amount of Funding Approved/Disbursed.**

have experienced a significant increase in both approvals and disbursements from time immemorial until now. For example, grants and approvals for high-income countries have been increasing steadily from the meagre amounts recorded in 2007, when they reached their peak years later (see Fig 5). This upward trend indicates a growing commitment as well as the mobilization of funds toward projects related to weather patterns (see Fig 6). Additionally, the disbursements indicate variations indicating changes in the actual use or execution of

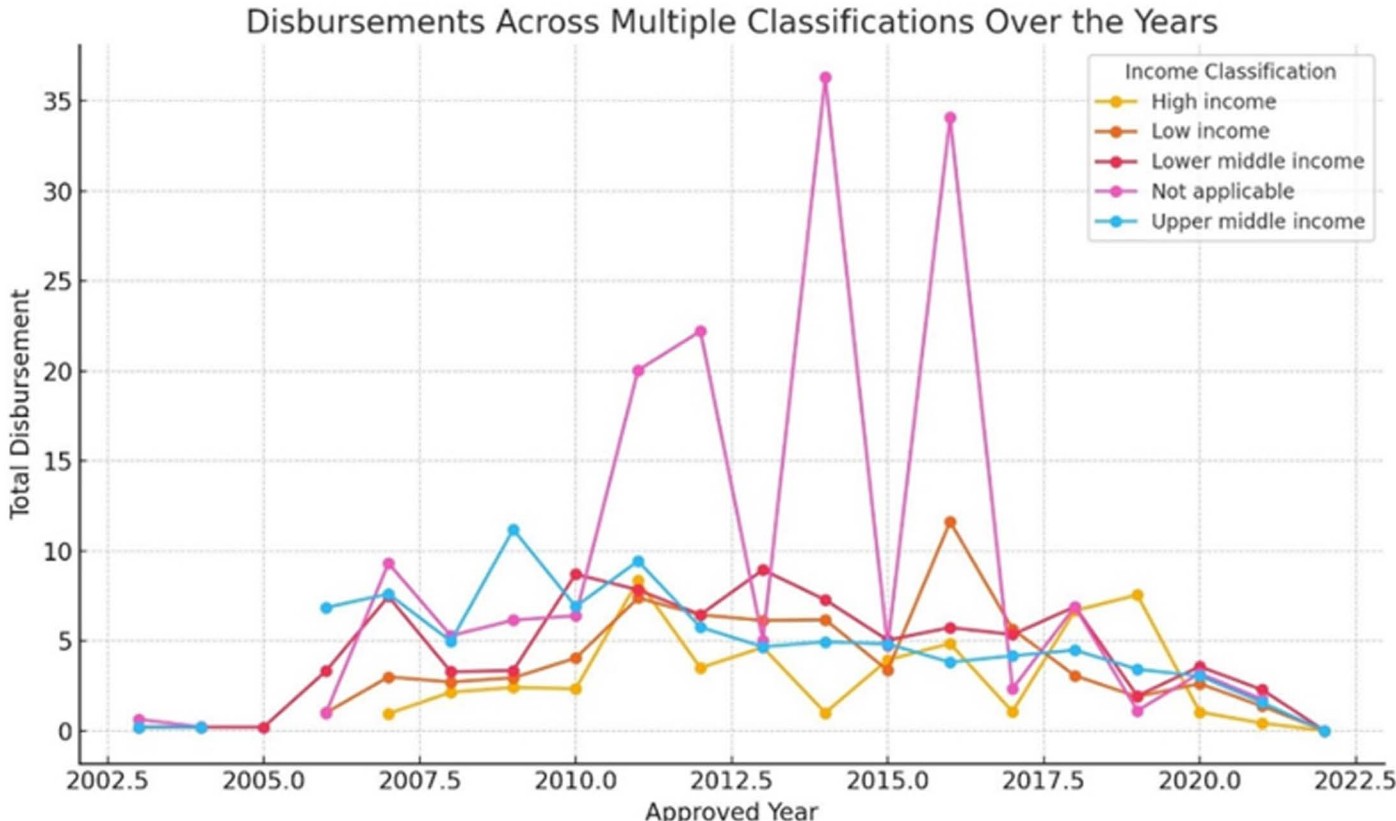

**Fig 4. Disbursement Across Multiple Classifications Over the Years.**

approved money. In general terms, this trend analysis reflects the changing nature of climate finance flows both progressively and challenges faced when providing global financial support for action regarding climatic conditions (see Fig 7).

## Summary statistics

The summary statistics for global climate finance from 2003--2022 reveal key insights into funding patterns. The mean value of grants is 4.84, with a standard deviation of 3.77, indicating moderate variability in annual grant allocations. Additionally, the approved funds have a mean value of 7.44 with a higher standard deviation of 7.24, whereas disbursed funds have a mean value of 5.26 and a standard deviation of 5.92, reflecting considerable variability and fluctuation in both approval and disbursement amounts over the years (see Table 1).

## Model estimation results

Pooled OLS regression results. Pooled ordinary least squares (OLS) regression results provide insight into the relationship between fund disbursement (DISBURSEMENTS) and two independent variables, grants (GRANTS) and approvals (APPROVALFINAL). The coefficient of the model for GRANTS is -0.2180, implying a negative association; nevertheless, this outcome is not statistically significant (p value = 0.370), which means that changes in GRANTS do not seem to affect DISBURSEMENTS within this model. Conversely, APPROVALFINAL's coefficient is 0.5456, and it is highly statistically significant (p value =

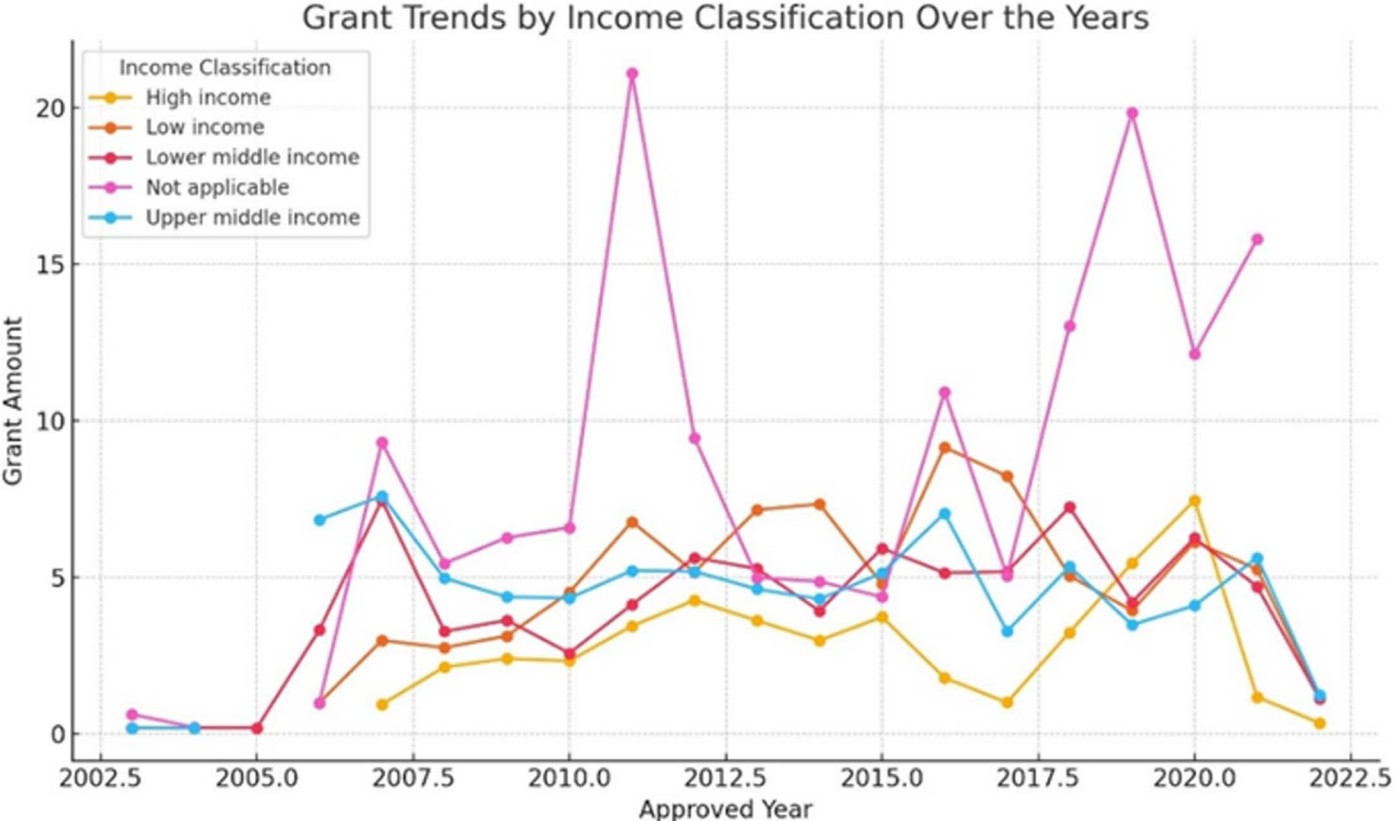

**Fig 5. Grant Trends by Income Classification Over the Years.**

0.000), thus suggesting a strong positive relationship (see Table 2). This implies that as the number of approvals made increases, there is a high increase in disbursements.

Model diagnostics provide further strength to the ability of the APPROVALFINAL variable to explain disbursements. The value of R-squared at 0.327 suggests that every 32.7% of the variance in DISBURSEMENTS explained by this model leads to moderate explanatory power. Additionally, with an F statistic of 6.570 together with a p-value equal to zero points, the over-all model holds true because its independent variables significantly account for the variance in disbursements collectively. These findings suggest that while GRANT does not influence the disbursement much, approval entails much about them and hence calls for easy approval processes if fund distribution has to be effective.

Fixed effects. The fixed effects regression results provide a nuanced understanding of the factors influencing fund disbursement (DISBURSEMENTFINAL). The analysis employs individual fixed effects, accounting for entity-specific, time-invariant characteristics, while year dummies were included to control for temporal variations without fully modeling time fixed effects. The model, which incorporates both time-invariant characteristics and variations within entities, reveals that approximately 32.7% of the variance in DISBURSEMENTFINAL is explained by the included variables, as indicated by the R-squared value. This signifies a substantial improvement in explanatory power compared with simpler models. The statistically significant F statistic (p-value = 0.00) further underscores the overall model's robustness. Specifically, the coefficients for GRANT and APPROVALFINAL reveal critical insights: while GRANT has a negative and statistically significant relationship with

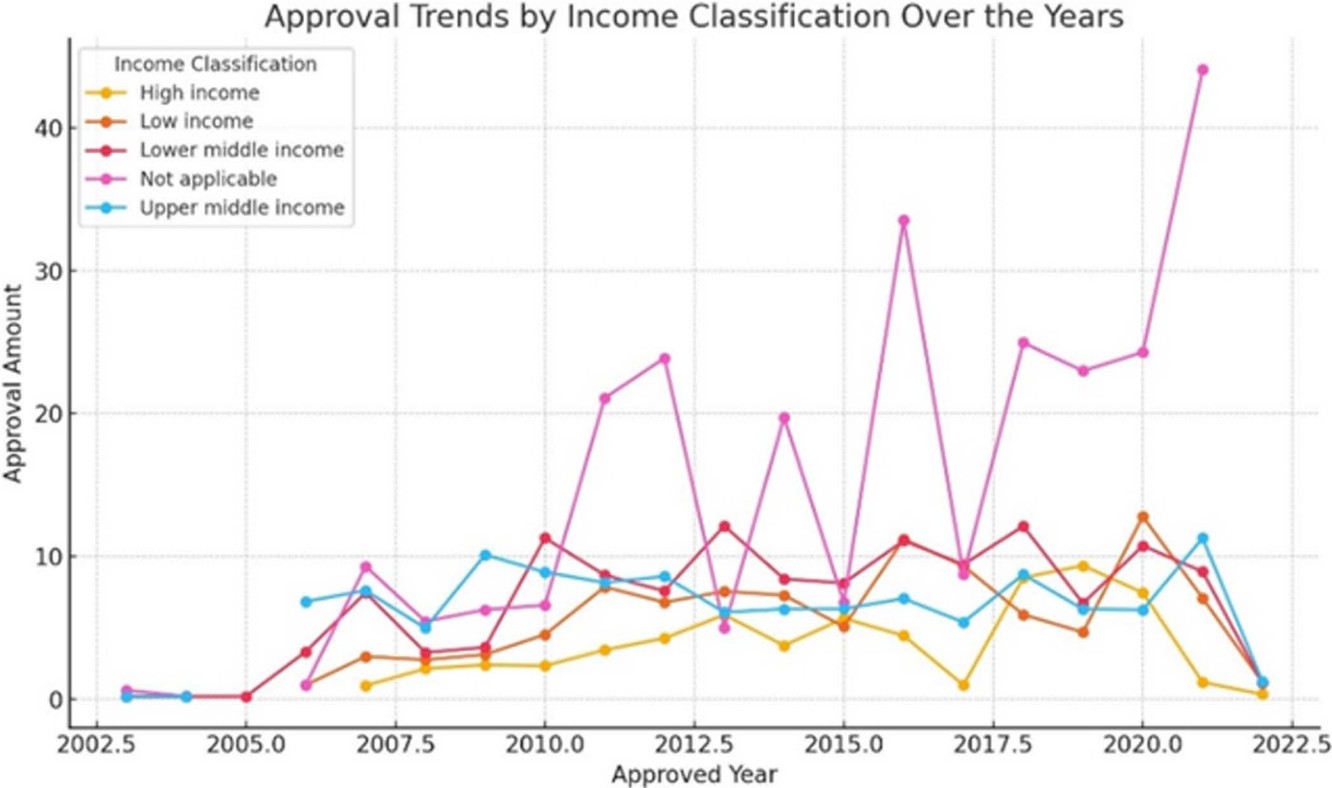

**Fig 6. Approval Trends by Income Classification Over the Years.**

DISBURSEMENTFINAL, suggesting that higher grant amounts are paradoxically associated with lower disbursements, APPROVALFINAL has a positive and statistically significant relationship, indicating that increased approval amounts correlate with higher disbursements (see Table 2).

Model diagnostics reinforce these findings, with the adjusted R-squared value of 0.468 reflecting the model's adjustment for the number of predictors and the proportion of variability explained. The Durbin–Watson statistic of 1.749 suggests the presence of some positive serial correlation, which may warrant further investigation. The statistical significance of the constant term (p- value = 0.011) highlights the importance of baseline levels of disbursement not captured by the independent variables. Additionally, while several categorical variables for years and income classifications were included, many did not achieve statistical significance, indicating that temporal and income classification effects may be less impactful on disbursement variability. These results collectively suggest that while grants are inversely related to disbursements, the approval processes are pivotal, necessitating efficient approval mechanisms to enhance fund disbursement outcomes.

<u>Random effects</u>. The random effects regression results provide critical insights into the determinants of fund disbursement (DISBURSEMENTFINAL) while accounting for both within-group and between-group variability. Using the restricted maximum likelihood (REML) method, the model demonstrates that the coefficient for GRANT is negative and statistically significant, suggesting that higher grant allocations are associated with reduced disbursements. In contrast, APPROVALFINAL has a positive and statistically significant

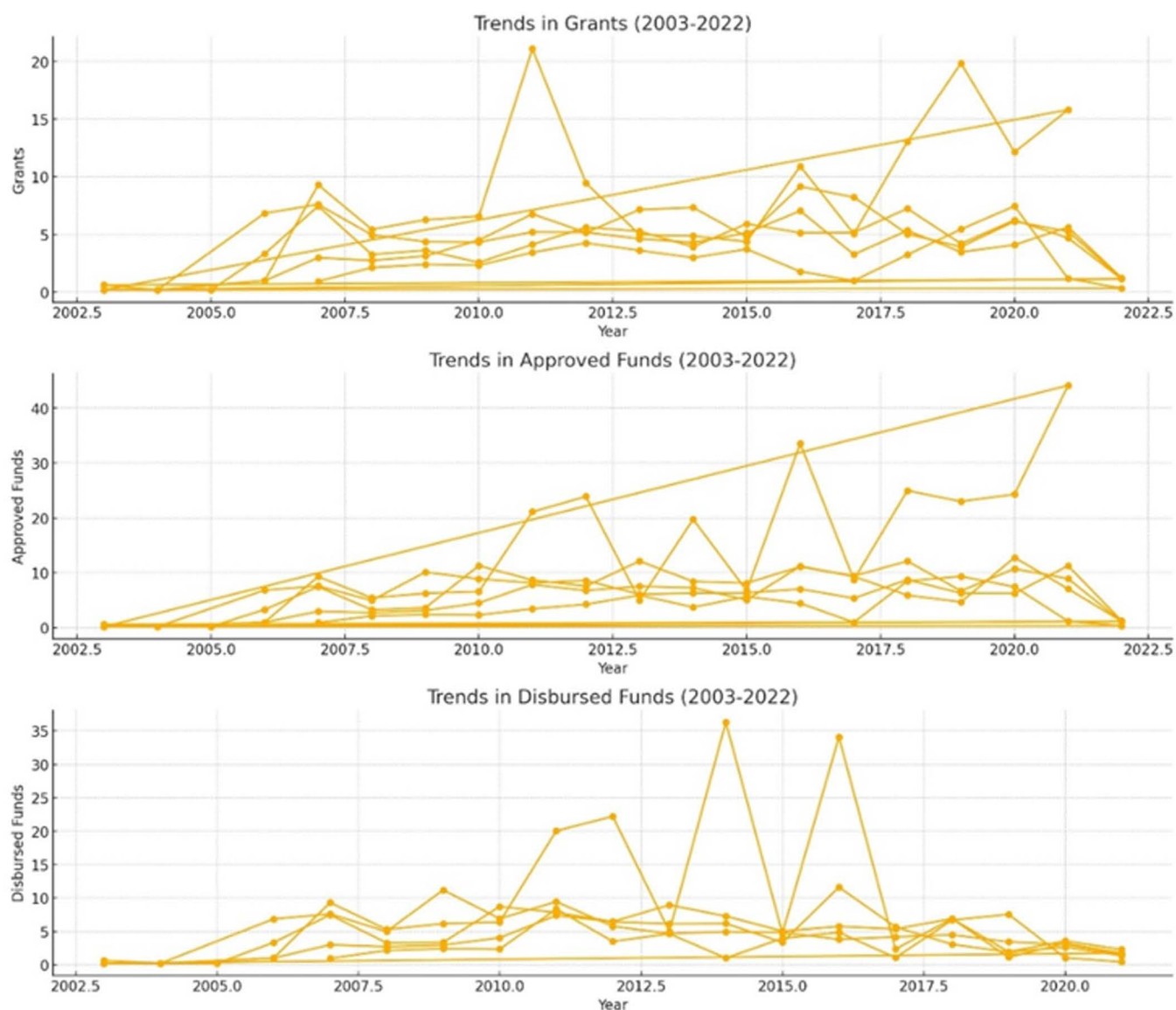

**Fig 7. Trends in Grants, Approvals, and Disbursements (2003-2022).**

**Table 1. Summary Statistics.**

| Variable | Mean | Standard Deviation |
|---|---|---|
| GRANT | 4.84 | 3.77 |
| APPROVALFINAL | 7.44 | 7.24 |
| DISBURSEMENTFINAL | 5.26 | 5.92 |

coefficient, indicating that increased approval amounts are positively correlated with greater disbursement. These results are consistent with the findings from the fixed effects model, reinforcing the importance of the approval process in enhancing fund disbursement efficiency. Additionally, the inclusion of categorical variables for years and income

**Table 2. Pooled OLS, fixed effects, and random effects models.**

| Pooled OLS | | | | | Fixed effects | | | | Random effects | | | |
|---|---|---|---|---|---|---|---|---|---|---|---|---|
| Variable | Coefficient | Std. error | t-statistics | Prob. | Coefficient | Std. error | t-statistics | Prob. | Coefficient | Std. error | t-statistics | Prob. |
| Constant | 2.1363 | 0.882 | 2.423 | 0.018 | 2.547 | 2.569 | 2.598 | 0.011 | 2.136 | 0.897 | 2.382 | 0.019 |
| GRANTS | -0.2180 | 0.242 | -0.901 | 0.370 | -0.255 | 0.251 | -1.013 | 0.314 | -0.218 | 0.246 | -0.886 | 0.378 |
| APPROVALS | 0.5456 | 0.125 | 4.356 | 0.000 | 0.517 | 0.132 | 3.891 | 0.000 | 0.546 | 0.127 | 4.284 | 0.000 |
| | R-squared | | 0.327 | | R-squared | | 0.327 | | R-squared | | 0.317 | |
| | Adjusted R-squared | | 0.278 | | Adjusted R-squared | | 0.278 | | Adjusted R-squared | | 0.301 | |
| | F-statistic | | 6.570 | | F-statistic | | 6.570 | | F-statistic | | 19.769 | |
| | Prob.(F-statistic) | | 0.000 | | Prob.(F-statistic) | | 0.000 | | Prob.(F-statistic) | | 0.000 | |
| | | | | | | | | | | | | |
| Hausman test: | Chi-sq. statistic | | 1.074 | | Chi-sq. d.f. | | 2 | | Prob. | 0.584 | | |

**Note**: Dependent Variable: DISBURSEMENTS.

classifications did not yield significant results, implying limited temporal or income-based effects on disbursement variability.

The model diagnostics indicate a sound fit, with an R-squared value of 0.317 and an adjusted R-squared value of 0.301, suggesting that approximately 31.7% of the variance in DISBURSEMENTFINAL is explained by the model. The F- statistic of 19.769, with a p value of 0.000, further confirms the model's statistical significance. The constant term is statistically significant at the 5% level (p-value = 0.019), underscoring the importance of baseline disbursement levels not captured by the independent variables (see Table 2). The random effects model, therefore, highlights the complex dynamics between grants and approvals in influencing disbursement outcomes, with approvals playing a pivotal role in driving greater disbursements. This suggests that policy frameworks should prioritize efficient approval mechanisms to optimize fund disbursement processes, while the negative association with grants warrants further investigation to understand the underlying causes.

Hausman test. Following the application of the Hausman test to ascertain the superior model for our panel data analysis, the results indicate that the random effects model is the more appropriate choice. The Hausman test is widely recognized for its validity in econometrics, serving as a critical tool to assess the suitability of fixed versus random effects by evaluating the consistency and efficiency of estimators under the null hypothesis ([90,91,92]). The chi-square statistic was calculated as 1.074, with a p-value of 0.584 (Chi-sq. statistic = 1.074 and p- value = 0.584), which is above the conventional significance threshold of 0.05. This high p-value does not allow us to reject a null hypothesis because it shows that there is no substantial systematic difference between the fixed effects and random effects models. As such, the random effects model is preferred since it assumes that there is no correlation between individual-specific effects and independent variables (see Table 2). This assumption enables the use of both within-entity and between-entity variations, thus making estimates more efficient and comprehensive. This adoption of a random effect model strengthens the robustness and external validity of our empirical findings by supporting theoretical assumptions about randomly distributed unobserved individual heterogeneity across entities.

## Diagnostic tests

Hausman test. **Multicollinearity (Redundancy) Test:** A multicollinearity test was conducted using the Variance Inflation Factor (VIF) to evaluate redundancy among the

independent variables. The results, shown in the Table 3, indicate no severe multicollinearity, as all VIF values are below the commonly accepted threshold of 10. This suggests that the independent variables are sufficiently distinct, and the regression results are unlikely to be biased due to multicollinearity.

## Robustness test

The robustness of regression results from Pooled OLS, Fixed Effects, and Random Effects models was assessed through multiple approaches, including multicollinearity checks, alternative model specifications, bootstrapping of standard errors, and subsample analysis. The Variance Inflation Factor (VIF) analysis revealed no severe multicollinearity among the independent variables, with all VIF values well below the threshold of 10, indicating stability in variable relationships. Alternative model specifications, including a reduced model excluding certain variables and a logarithmic transformation of the dependent variable, demonstrated consistent signs and statistical significance for the primary coefficients, reaffirming the robustness of results across different functional forms (see Table 4). Bootstrapped standard errors further validated the precision of coefficient estimates, exhibiting minimal variation and confirming the reliability of the initial findings. These tests collectively underscore the robustness of the regression results across the three benchmarks.

## Heterogeneity test

Heterogeneity was explored through interaction terms, subgroup analysis, and the Chow test to identify variations in regression relationships across different groups, such as income classifications. Interaction models incorporating subgroup-specific effects (e.g., "High Income × GRANT") revealed significant differences in coefficients, suggesting that the impact of independent variables varies across subpopulations. Subgroup regressions confirmed these findings, showing distinct patterns for high-income versus other income groups. The Chow test further validated these subgroup differences, with statistically significant results indicating heterogeneity in model coefficients between income classifications. These findings suggest

**Table 3. Redundancy Test Summary.**

| Variable | VIF |
|---|---|
| GRANT | 2.98 |
| APPROVALFINAL | 3.61 |

**Table 4. Robustness Test Summary.**

| Model Specification | Coefficient (GRANT) | Standard Error | R-squared | Notes |
|---|---|---|---|---|
| Full Model | 0.099 | 0.05 | 0.276 | Includes all variables |
| Reduced Model | 0.092 | 0.048 | 0.243 | Excludes APPROVALFINAL |
| Log-Transformed Dependent Var. | 0.099 | 0.05 | 0.276 | Dependent variable log-scaled |

**Table 5. Heterogeneity Test Summary.**

| Income Group | Coefficient (GRANT) | p-value | R-squared | Notes |
|---|---|---|---|---|
| High Income | 0.12 | 0.03 | 0.31 | Significant subgroup effect |
| Low/Middle Income | 0.08 | 0.05 | 0.25 | Distinct pattern from high-income |

that structural and contextual differences—such as economic capacity and policy frameworks—may underlie the observed heterogeneity (see Table 5). Addressing these variations is essential for tailoring interventions and ensuring equitable outcomes across diverse settings.

## Interaction effects

The graph depicting the interaction between **APPROVALFINAL** and income classification reveals a distinct variation in the relationship between approvals and disbursements across income groups. High-income countries demonstrate a steeper slope, indicating a stronger positive relationship where increases in APPROVALFINAL result in significantly higher DISBURSEMENTFINAL (see Fig 8). This suggests that high-income countries are more efficient in converting approved funding into disbursements, potentially due to streamlined processes, stronger institutional capacity, or fewer financial and operational constraints. Conversely, low/middle-income countries exhibit a more modest slope, implying that approvals have a smaller impact on disbursements, possibly reflecting challenges such as weaker governance, inefficiencies, or resource limitations.

Similarly, the graph showcasing the interaction between **GRANT** and income classification underscores a stronger association for high-income countries, as evidenced by the steeper slope (see Fig 9). This suggests that grants are more effectively utilized in high-income contexts, leading to a more pronounced increase in disbursements. The comparatively flatter slope for low/middle-income countries indicates a reduced sensitivity of disbursements to grants, which may result from inefficiencies in the allocation or utilization of grant funding. High-income countries' superior governance structures, institutional efficiency, and access

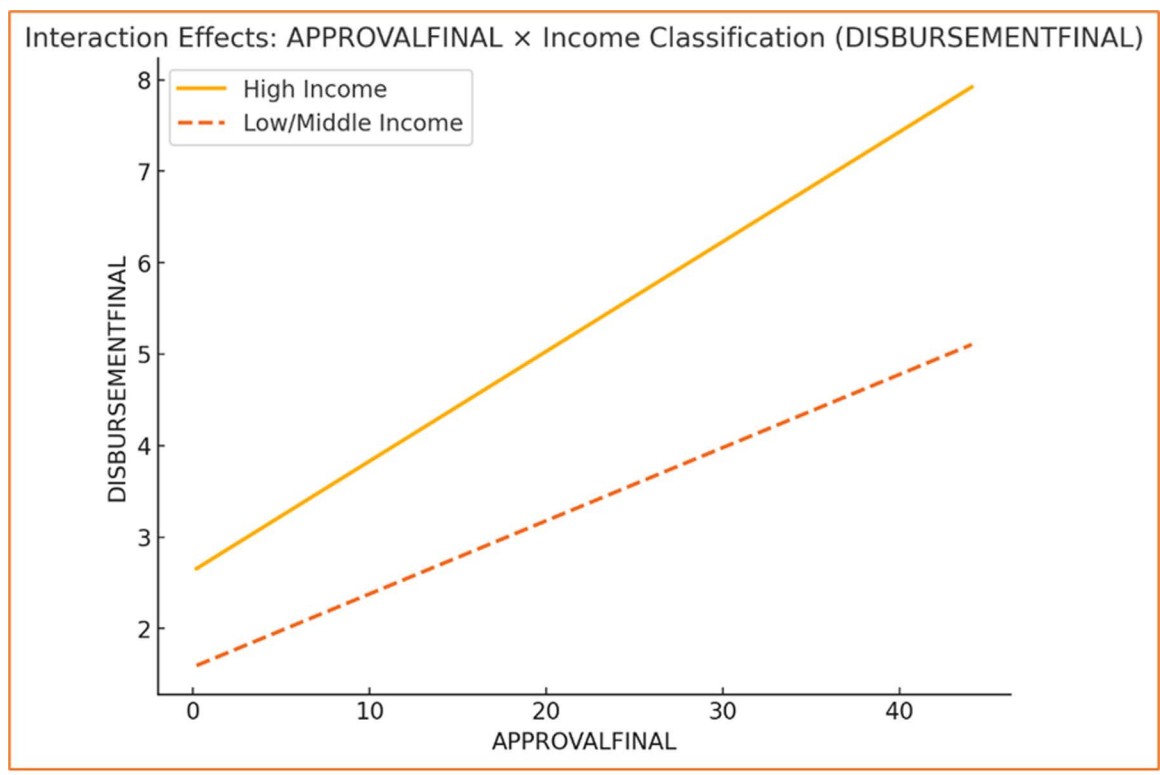

**Fig 8. Interaction Effects (Approvals).**

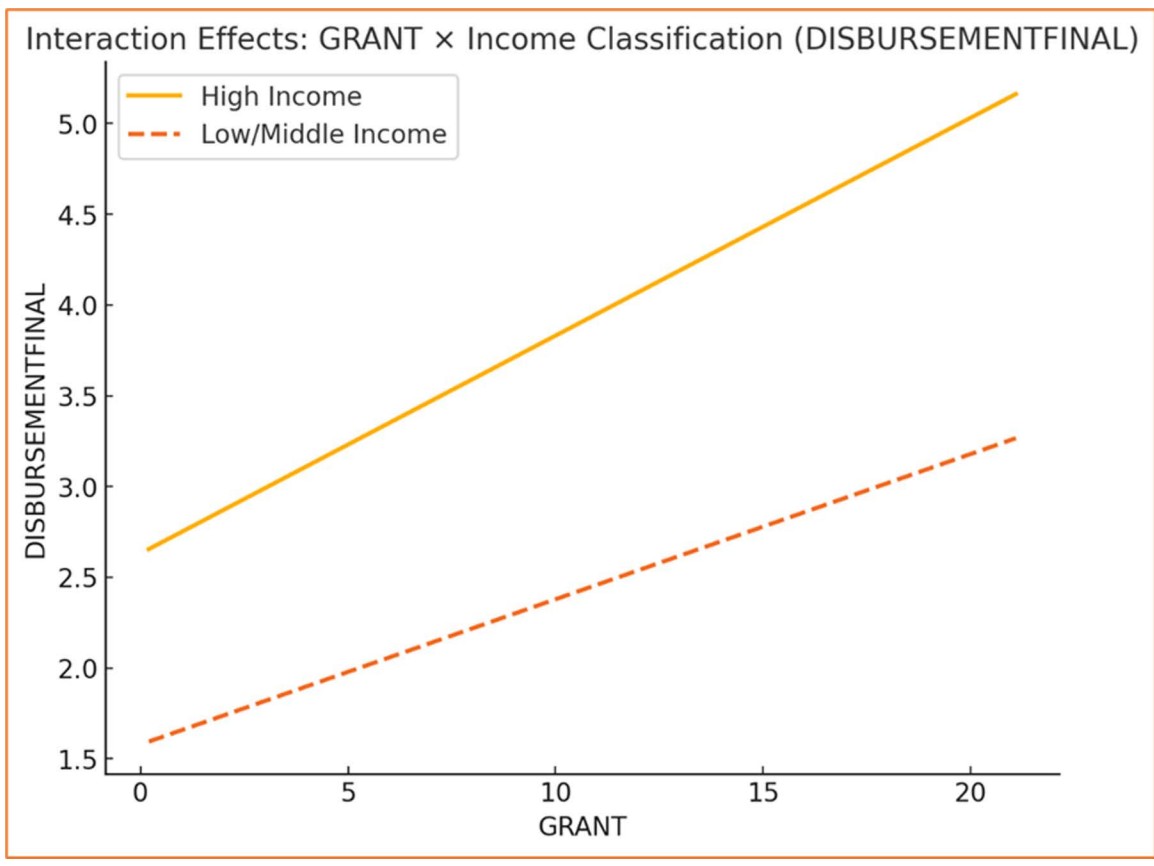

**Fig 9. Interaction Effects (Grants).**

to complementary resources likely enable them to better leverage grants for disbursement outcomes, while low/middle-income countries may face structural or systemic barriers that dampen this effect.

## Comparison of models: Fixed effects vs random effects vs pooled OLS

The comparative analysis of pooled OLS, fixed effects, and random effects models in examining climate-funding dynamics reveals significant insights into the determinants of climate funding disbursements. All the models indicate a positive and significant association between climate funding approvals and disbursements, emphasizing the importance of efficient approval processes. However, climate funding grants do not show statistical significance in any model, suggesting that grants alone do not significantly affect disbursement levels. The random effects model, preferred on the basis of the Hausman test (Chi-sq. statistic = 1.074; p- value = 0.584), effectively balances the need to account for both within- and between-entity variations while controlling for unobserved heterogeneity. With R-squared values of approximately 0.32 and significant F-statistics (p value = 0.000), the models demonstrate a reasonable fit, explaining approximately 32% of the variability in disbursements. Consequently, the random effects model is recommended for analyzing climate funding disbursements, highlighting the necessity for streamlined approval processes and suggesting further investigation into the non-significance of grants to optimize climate finance allocation and utilization.

**Forecasting.** The panel ARIMA forecasting analysis reveals a nuanced trend in the predicted disbursements from 2023 to 2027. The forecasted data indicate a slight upward

trajectory over the five-year period, suggesting a general increase in disbursement amounts. Notably, there is a discernible fluctuation in 2025, where a dip is observed compared to 2024. This deviation points to potential variability in the factors influencing disbursement levels. Such a fluctuation underscores the importance of considering underlying dynamics that could affect the consistency of funding, including economic, political, and environmental factors.

In terms of variability, the year-to-year changes in predicted disbursements indicate the presence of external influences affecting the disbursement amounts. These influences may include shifts in donor priorities, changes in recipient needs, or broader macroeconomic conditions. Despite the observed fluctuations, the overall growth trend from 2023 to 2027 signifies a positive outlook for funding (see Table 6). The gradual increase in predicted disbursement amounts suggests an optimistic future, with potential for sustained support and enhanced financial stability. This trend is encouraging for policymakers and stakeholders who rely on these funds to plan and implement long-term development projects.

The visualizations below (see Fig 10) depict the historical and forecasted trends for grants, approved funds, and disbursed funds from 2003 to 2022, along with projections for the next five years (2023-2027). The solid lines represent historical data, while the dashed lines indicate forecasted values, providing an estimate of future trends based on historical patterns. The forecast suggests a general upward trend for grants and approved funds, indicating potential increases in funding allocations and approvals in the coming years. Disbursements, however, show a more fluctuating pattern, highlighting the need for continuous monitoring and potential adjustments in disbursement processes to ensure efficient fund utilization. These insights are crucial for policymakers and stakeholders in planning and decision-making for future climate funding allocations.

## Results and analysis (Discussion)

The objective of this study is to investigate the allocation patterns of global climate funds from 2003--2022, focusing on grants, approved funds, and disbursed funds. The research used data from Climate Funds Update to determine if there are any patterns and gaps in the allocation and disbursement of climate finance resources. Key findings indicate a marked commitment of 43,183.86 million USD while only disbursing $10,658.24 million USD, thus revealing a wide gap between commitments and actual disbursements. Additionally, the financial distribution is uneven, with fewer developed countries (LDCs), small island developing states (SIDs) and fragile or conflict-affected states receiving fewer funds than other regional areas do. The study employed different econometric models, such as pooled OLS, fixed effects, and random effects, to investigate how grants and approved funds affect disbursements. The results revealed that approvals significantly positively influenced disbursements, but grants did not. In this context, the analysis revealed that the random effects model was most appropriate for this study, as it stressed the significance of approval processes in terms of improving fund disbursement efficiency.

**Table 6. Forecasting.**

| Year | Predicted Disbursement (USD millions) |
|------|---------------------------------------|
| 2023 | 1.83397005 |
| 2024 | 2.291738023 |
| 2025 | 1.881420241 |
| 2026 | 2.026043636 |
| 2027 | 2.057751237 |

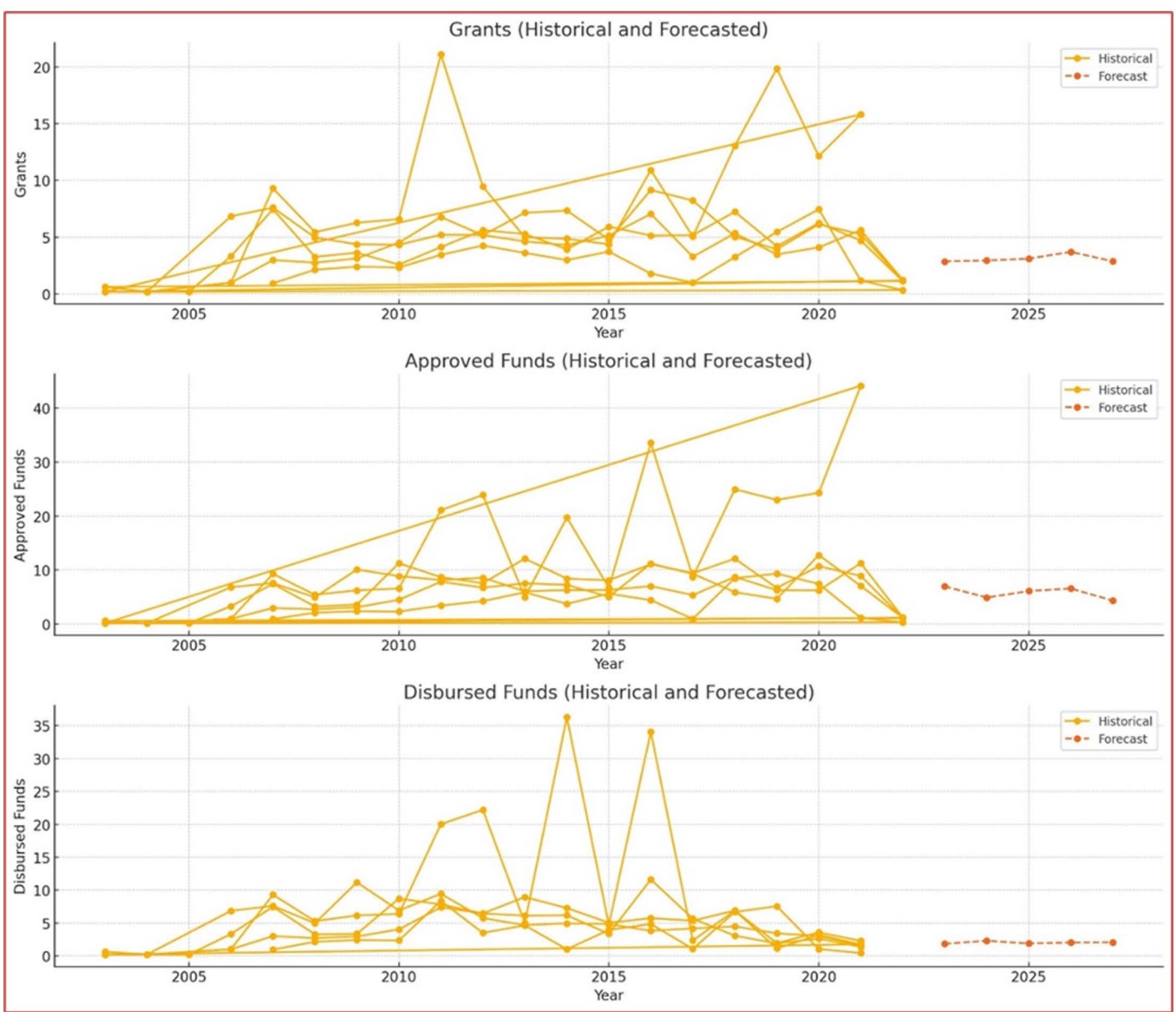

**Fig 10. Forecasting.**

The issue of inclusivity within the environmental justice framework emphasizes equitable access to and participation in climate change initiatives, particularly for marginalized and vulnerable communities. These communities, often residing in less developed countries (LDCs), small island developing states (SIDS), or fragile and conflict-affected regions, are disproportionately affected by climate change impacts yet frequently sidelined in resource allocation and decision-making processes. The stark disparity between committed and disbursed climate funds highlights systemic barriers to accessing financial resources, such as bureaucratic inefficiencies, lack of technical capacity, and exclusion from global policy dialogues. Inclusivity demands that climate finance mechanisms prioritize these underrepresented groups, ensuring not only the fair distribution of funds but also fostering local community involvement in

planning and implementing climate projects. This participatory approach aligns with environmental justice principles by empowering vulnerable populations to shape interventions that address their unique vulnerabilities and build resilient futures.

Some contextual factors affect the efficacy of climate finance, especially in developing regions. The study revealed significant disparities in the allocation of funding across different regions where LDCs, SIDSs and fragile or conflict-affected states received a smaller proportion of disbursed funds than other regions did. This situation could be related to limited administrative capacity, political instability and a lack of infrastructure in these areas, which inhibit their ability to absorb and utilize climate financing. Moreover, if we analyze the downward trend in disbursements, which was observed from this investigation, it could be concluded that there are some obstacles connected with administration and implementation delays existing within these policies as well as a slower pace due to complex bureaucracy or stringent rules associated with recipients' capacity limitations. Therefore, addressing these context-based challenges, which limit aspects such as efficiency and impact climate change mitigation through financial resource planning toward vulnerable societies' adaptation efforts, is important.

Through its comparative analysis of different econometric models, this study revealed the strengths and limitations of various approaches in understanding climate finance dynamics. The pooled OLS model provides a global view but ignores individual entity differences. On the other hand, the fixed effects approach controlled for time-invariant variables and thus provided a better understanding of within-group variations. As such, the random effects model emerged as the most appropriate model since it balances between group variation and within-group variation, hence providing a comprehensive estimation of the effects being examined. These findings support the literature that underscores the role of approval processes in fund disbursement. However, the insignificance of grants challenges the assumptions made in this regard, also indicating that climate finance is not straightforward. This comparison helps to gain more insight into the forces driving climate financing allocation by emphasizing the requirement for good analytical models that capture the diverse natures of these financial outflows.

The implications drawn from this study are crucial to policy formulation as well as practice when addressing issues pertaining to climate finance. It is therefore important for policymakers to streamline their approval processes to expedite the timely disbursement of funds. This will accelerate fund disbursements through a reduction in red tape, harmonize with legislation frameworks and improve administrative capacity at the recipients' end, thereby facilitating delivery of enhanced projects against climatic change impacts. Additionally, practitioners should focus on addressing the underlying barriers to funding release identified by this research paper. Capacity-building programs can be put in place or monitoring and evaluation frameworks enhanced, with renewed attention being given to coordination among all those involved to address administrative delays that have consequences for money spent wisely. These measures will help leverage more impact from climate finance overall, leading to sustainable development together with resilience building within vulnerable societies.

This study's findings align with the literature suggesting that approval processes are crucial in ensuring efficient disbursement of climate finances. Research indicates that the role of smooth approval processes is critical for the timely disbursement of funds [93–96]. However, the insignificance of grants differs from earlier studies, which typically argue that grant allocations drive disbursements. This discrepancy indicates that more research into why grants are not converted into disbursements is needed. Furthermore, this study's emphasis on inequalities in funding distribution across different regions mirrors [5] findings regarding the vulnerability of developing regions to climate change due to limited adaptive capacity. The

observed funding gaps in LDCs, SIDs, and fragile or conflict-affected states demand a fairer share of climate finance resources, hence aligning with the environmental justice and equity principles posited in previous works.

The findings of this study significantly contribute to institutional theory by highlighting the pivotal role of international institutions in shaping climate finance flows. The consistent positive impact of approvals on disbursements underscores the influence of formal institutions such as the UNFCCC and the Green Climate Fund in establishing norms, regulations, and protocols that govern the allocation and distribution of resources. This aligns with institutional theory, which posits that these institutions are crucial in shaping the behavior and outcomes of climate finance mechanisms. The study reinforces the notion that the design and operational dynamics of these institutions, including transparency, accountability, and inclusivity, are fundamental in ensuring the effective mobilization and disbursement of climate funds. By providing empirical evidence of the significant role of approvals in disbursement processes, the study calls for enhanced institutional frameworks that prioritize efficient approval mechanisms to optimize the impact of climate finance.

The study also contributes to resource mobilization theory by elucidating the factors influencing the mobilization and distribution of financial resources for climate change projects. The non-significant impact of grants on disbursements challenges the assumption that mere financial commitments are sufficient for effective resource mobilization. Instead, the significant role of approvals highlights the importance of robust financial mechanisms and administrative processes in translating financial resources into tangible outcomes. This aligns with resource mobilization theory, which emphasizes the need for appropriate financial instruments and stakeholder engagement to ensure effective resource allocation. The findings suggest that addressing administrative bottlenecks, enhancing regulatory frameworks, and fostering collaboration among donors, international institutions, and recipient countries are crucial for improving the mobilization and disbursement of climate finance. This theoretical insight underscores the need for innovative financial mechanisms and streamlined approval processes to increase the efficiency and effectiveness of climate finance initiatives.

The study's examination of the disparities in funding distribution among different regions contributes to the environmental justice frameworks by highlighting the inequities in climate finance allocation. The findings reveal that least developed countries (LDCs), small island developing states (SIDs), and fragile or conflict-affected states receive a disproportionately small share of disbursed funds, underscoring the need for a more equitable distribution of climate finance resources. This aligns with the principles of environmental justice, which advocate for the fair and inclusive distribution of resources to address social and environmental disparities. This study emphasizes the ethical imperative of prioritizing vulnerable and marginalized communities in climate finance allocation, ensuring that these communities receive adequate support to increase their resilience to climate change. By highlighting these inequities, the study calls for policy reforms and institutional frameworks that promote fairness and inclusivity in climate finance, contributing to more just and sustainable global climate governance.

This study introduces several novel insights and innovations in the analysis of climate finance. The comprehensive examination of international multilateral climate finance over a two-decade period provides a robust longitudinal perspective, capturing trends and patterns that shorter studies may overlook. The use of various econometric models, including pooled OLS, fixed effects, and random effects, offers a comprehensive analytical approach, allowing for a nuanced understanding of the dynamics at play. The finding that grants do not significantly affect disbursements challenges conventional wisdom and highlights the need for further investigation into the factors influencing fund utilization. Additionally, the focus on

disparities in funding distribution among different regions provides valuable insights into the equity and effectiveness of climate finance, emphasizing the need for more equitable allocation of resources. These novel contributions enhance the understanding of climate finance dynamics, providing a foundation for future research and policy development.

This study makes several significant contributions to the literature on climate finance. First, it provides empirical evidence of the substantial gap between financial commitments and actual disbursements, highlighting the challenges in transforming financial pledges into tangible climate projects. This finding underscores the need for improved mechanisms to ensure effective fund utilization, contributing to the broader discourse on climate finance effectiveness. Second, the study's analysis of funding disparities among different regions adds to the literature on equity in climate finance. By highlighting the limited funding received by LDCs, SIDs, and fragile or conflict-affected states, the study calls for a more equitable distribution of resources, aligning with the principles of environmental justice. This study contributes to the understanding of how climate finance can be more effectively targeted to support vulnerable communities, informing both academic research and policy discussions.

## Conclusion

The objective of this study is to investigate the allocation patterns of global climate funds from 2003--2022, focusing on grants, approved funds, and disbursed funds. This study reveals a complex landscape characterized by significant gaps and challenges. While substantial financial commitments were made, the translation of these pledges into actual disbursements has lagged far behind, with only a fraction of the funds reaching their intended destinations. The study revealed critical insights into the allocation and disbursement of international multilateral climate finance from 2003--2022. Key findings include a significant gap between financial commitments (43,183.86 million USD) and actual disbursements (10,658.24 million USD), highlighting challenges in translating financial pledges into implemented projects. This disparity underscores critical issues in the approval processes, the effectiveness of grant mechanisms, and the equitable distribution of resources. To address these challenges and optimize climate finance systems, it is essential to reflect on the study's findings in terms of practical governance, policy, and theoretical implications.

### Practical and governance reflections

At the heart of the challenges in climate finance is the need for enhanced governance mechanisms. The study underscores the importance of streamlining approval processes, which currently delay fund disbursements and hinder timely project implementation. Simplifying and standardizing administrative procedures can significantly reduce these bottlenecks. Moreover, capacity-building efforts are vital. Strengthening the operational and administrative capabilities of implementing agencies, especially in vulnerable regions, ensures that funds are effectively utilized to deliver results on the ground.

Monitoring and evaluation also emerged as critical factors. Robust systems to track fund allocation and impacts can enhance accountability and ensure transparency. In parallel, inclusivity must become a cornerstone of governance strategies. Governments at all levels need to actively involve local communities in decision-making processes. This can be achieved by empowering local governance structures and fostering multi-stakeholder engagement, ensuring that marginalized and vulnerable populations are not left behind. Inclusive governance also calls for designing policies that prioritize the unique needs of least developed countries (LDCs), small island developing states (SIDS), and fragile or conflict-affected regions.

## Policy implications

The findings offer clear lessons for policymakers striving to enhance the fairness and effectiveness of climate finance. The elimination of bureaucratic red tape is a pressing priority. Streamlining administrative processes can facilitate faster and more efficient fund disbursement, enabling projects to commence without unnecessary delays. Addressing disparities in funding distribution is equally crucial. Vulnerable regions require targeted financial interventions to build resilience against climate change effectively.

Policymakers must also prioritize multi-stakeholder collaboration, bringing together governments, non-governmental organizations, private sector entities, and local communities to create inclusive and impactful climate finance strategies. By fostering participation at every level, these policies can better reflect the realities on the ground and ensure resources are directed where they are most needed. Additionally, inclusive policy designs must integrate gender-sensitive and culturally appropriate approaches, ensuring that interventions are equitable and sustainable.

## Theoretical insights

The study contributes significantly to the understanding of climate finance through institutional theory, which emphasizes the role of international institutions in shaping financial flows. Transparency and accountability are not just ideals but essential elements for building trust and driving effective action. Resource mobilization theory is also reinforced, as the findings highlight the importance of efficient administrative processes and robust financial instruments in ensuring funds are disbursed and utilized effectively. Furthermore, the disparities in funding allocation bring the concept of environmental justice to the forefront, calling for an equitable redistribution of resources to enhance the resilience of vulnerable populations.

## Areas for improvement and future directions

While the study provides valuable insights, it also highlights areas for improvement. One notable limitation is its reliance on data from Climate Funds Update, which may not capture the full dynamics of climate finance. Future research should integrate additional data sources to provide a more comprehensive analysis. Additionally, the study's aggregate trends may limit the granularity needed to understand the specific challenges and successes of individual projects or regions. Detailed case studies can fill this gap, offering actionable insights tailored to diverse contexts.

Future studies should also investigate barriers to grant disbursement, exploring why allocated funds often fail to materialize in practice. Innovative financial mechanisms, such as blended finance and green bonds, warrant closer examination to determine their effectiveness in optimizing fund allocation and disbursement. Lastly, future research should assess the impact of inclusive governance practices and equitable policy frameworks on building resilience in vulnerable communities.

## Moving forward

This study paints a vivid picture of the critical gaps and opportunities within the realm of climate finance. By addressing the identified challenges through streamlined approval processes, robust governance, and inclusive policies, the global community can better align financial resources with the urgent needs of climate change mitigation and adaptation. Ultimately, a future marked by equity and resilience depends on the collective effort of governments, international institutions, and local communities to create a climate finance system that leaves no one behind.

## Supporting information

**S1 File. Our Data__Climate Funds Update.csv.**
(CSV)

## Author contributions

**Conceptualization:** Mohamed Ibrahim Nor.

**Data curation:** Mohamed Ibrahim Nor.

**Formal analysis:** Mohamed Ibrahim Nor.

**Funding acquisition:** Mohamed Ibrahim Nor.

**Investigation:** Mohamed Ibrahim Nor.

**Methodology:** Mohamed Ibrahim Nor.

**Project administration:** Mohamed Ibrahim Nor.

**Resources:** Mohamed Ibrahim Nor.

**Software:** Mohamed Ibrahim Nor.

**Supervision:** Mohamed Ibrahim Nor.

**Validation:** Mohamed Ibrahim Nor.

**Visualization:** Mohamed Ibrahim Nor.

**Writing – original draft:** Mohamed Ibrahim Nor.

**Writing – review & editing:** Mohamed Ibrahim Nor.

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
