## [Decision Letter · Decision Letter 0]

17 Oct 2024

PONE-D-24-32451Investigating the Dynamics of Climate Finance Disbursements: A Panel Data Approach from 2003 to 2022PLOS ONE

Dear Dr. Nor,

Thank you for submitting your manuscript to PLOS ONE. After careful consideration, we feel that it has merit but does not fully meet PLOS ONE’s publication criteria as it currently stands. Therefore, we invite you to submit a revised version of the manuscript that addresses the points raised during the review process.

If applicable, we recommend that you deposit your laboratory protocols in protocols.io to enhance the reproducibility of your results. Protocols.io assigns your protocol its own identifier (DOI) so that it can be cited independently in the future. For instructions see: https://journals.plos.org/plosone/s/submission-guidelines#loc-laboratory-protocols [https://c05y1x9s.r.us-east-2.awstrack.me/L0/https:%2F%2Fjournals.plos.org%2Fplosone%2Fs%2Fsubmission-guidelines%23loc-laboratory-protocols/1/010f0191fece1910-218e346b-f190-4c7a-8b76-fd70cad2da64-000000/4o0OP5yvL4o-2ahUqEHJlurltBs=17] .

Additionally, PLOS ONE offers an option for publishing peer-reviewed Lab Protocol articles, which describe protocols hosted on protocols.io. Read more information on sharing protocols at  https://plos.org/protocols?utm_medium=editorial-email&utm_source=authorletters&utm_campaign=protocols [https://c05y1x9s.r.us-east-2.awstrack.me/L0/https:%2F%2Fplos.org%2Fprotocols%3Futm_medium=editorial-email%26utm_source=authorletters%26utm_campaign=protocols/1/010f0191fece1910-218e346b-f190-4c7a-8b76-fd70cad2da64-000000/gOvLT3C65Q01X_kLQr3xSs9OG80=17 ]

We look forward to receiving your revised manuscript.

Kind regards,

Xiaoxuan Kao

Guest Editor

PLOS ONE

Journal Requirements:

1. When submitting your revision, we need you to address these additional requirements. Please ensure that your manuscript meets PLOS ONE's style requirements, including those for file naming. The PLOS ONE style templates can be found at https://journals.plos.org/plosone/s/file?id=wjVg/PLOSOne_formatting_sample_main_body.pdf and https://journals.plos.org/plosone/s/file?id=ba62/PLOSOne_formatting_sample_title_authors_affiliations.pdf 2. Thank you for stating the following financial disclosure: "Center for Research and Development (CRD), SIMAD University." Please state what role the funders took in the study.  If the funders had no role, please state: "The funders had no role in study design, data collection and analysis, decision to publish, or preparation of the manuscript." If this statement is not correct you must amend it as needed. Please include this amended Role of Funder statement in your cover letter; we will change the online submission form on your behalf. 3. Please provide a complete Data Availability Statement in the submission form, ensuring you include all necessary access information. If your research concerns only data provided within your submission, please write "All data are in the manuscript and/or supporting information files" as your Data Availability Statement.

Reviewers' comments:

Reviewer's Responses to Questions

**Comments to the Author**

1. Is the manuscript technically sound, and do the data support the conclusions?

Reviewer #1: Partly

Reviewer #2: Yes

2. Has the statistical analysis been performed appropriately and rigorously? 

Reviewer #1: Yes

Reviewer #2: Yes

3. Have the authors made all data underlying the findings in their manuscript fully available?

Reviewer #1: Yes

Reviewer #2: Yes

4. Is the manuscript presented in an intelligible fashion and written in standard English?

Reviewer #1: Yes

Reviewer #2: Yes

5. Review Comments to the Author

Reviewer #1: 1. The Introduction section lacks organization, and it is recommended to extract key information;

2. Please provide a comprehensive and in-depth review of the literature review in order to extract key arguments, reveal the current status and trends in the research field, and provide a solid theoretical foundation and direction for subsequent research.

3. Fixed effects are used to study the impact of core variables on the dependent variable. When using fixed effects, please indicate whether they are bidirectional fixed or time or individual fixed.

4. The Hausman test can cite research to prove the validity of this test.

5. After the regression results of the three benchmarks, it is recommended to conduct a robustness test and heterogeneity test to test whether the regression results are robust and analyze the reasons for the existence of heterogeneity.

6. Lack of innovation, the entire research process is relatively simple, including whether the data has multicollinearity and how to solve multicollinearity, which was not mentioned in the article. Please supplement.

Reviewer #2: COMMENTS:

1) Create specific section for METHOD and organize into sub-sections as follows (1) Type and Source of Data (example, this study employs largely secondary data....) (2) Research Method and Model Specifications etc.

2) Change Discussion section to RESULTS AND ANALYSIS (DISCUSSION) - Must tally with the aims/purposes of the study - Please explain more on the issue of inclusivity (environmental justice framework) which involved community access and participation in climate change initiatives.

3) So many IN-TEXT CITATIONS but not addressed in the References section

4) Most of the FIGURES and TABLE do not have their sources cited.

5) Some REFERENCES are not formatted properly and have missing details (year, pages, publisher, DOI)

6) The CONCLUSION of the study is somewhat disorganized, especially regarding the implications. It would be more effective to structure it into distinct sub-sections, such as Practical/Governance Implications, Policy Implications, and Theoretical Implications. Please explain how could the issue of inclusivity i.e community participation and equitable access to climate change initiatives can be addressed by governments at all levels. Several key concepts can also be included in this issues notably multi-stakeholder engagement, empowering local governance, and implementing inclusive policy designs that prioritize vulnerable populations. . Please provide some best practices and suggestions for improvement in this study (as stated earlier as the aims of this study). Based on the limitations of the study, please add suggestion for Future Studies.

7) The major reference from which the DATASET was obtained, notably the Climate Funds Update (CFU), is not properly cited in the References section at least please provide the website link/DOI link (if it's retrieved from jounals or official documents)

8) It is stated in the ABSTRACT "The FORECASTING results indicate a positive trend in disbursements from 2023 to 2027, with potential fluctuations driven by external factors, but missing reporting in the content.

6. PLOS authors have the option to publish the peer review history of their article (what does this mean? ). If published, this will include your full peer review and any attached files.

**Do you want your identity to be public for this peer review?** For information about this choice, including consent withdrawal, please see our Privacy Policy .

Reviewer #1: No

Reviewer #2: No

---

## [Author Response · Author response to Decision Letter 1]

2 Dec 2024

We have uploaded two rebuttal letters for your action.

---

## [Decision Letter · Decision Letter 1]

12 Jan 2025

Investigating the Dynamics of Climate Finance Disbursements: A Panel Data Approach from 2003 to 2022

PONE-D-24-32451R1

Dear Dr. NOR,

We’re pleased to inform you that your manuscript has been judged scientifically suitable for publication and will be formally accepted for publication once it meets all outstanding technical requirements.

Kind regards,

Xiaoxuan Kao

Guest Editor

PLOS ONE

Additional Editor Comments (optional):

Reviewers' comments:

Reviewer's Responses to Questions

**Comments to the Author**

1. If the authors have adequately addressed your comments raised in a previous round of review and you feel that this manuscript is now acceptable for publication, you may indicate that here to bypass the “Comments to the Author” section, enter your conflict of interest statement in the “Confidential to Editor” section, and submit your "Accept" recommendation.

Reviewer #2: All comments have been addressed

2. Is the manuscript technically sound, and do the data support the conclusions?

Reviewer #2: Yes

3. Has the statistical analysis been performed appropriately and rigorously? 

Reviewer #2: N/A

4. Have the authors made all data underlying the findings in their manuscript fully available?

Reviewer #2: Yes

5. Is the manuscript presented in an intelligible fashion and written in standard English?

Reviewer #2: Yes

6. Review Comments to the Author

Reviewer #2: I am fully satisfied with the revisions made by the author(s). Each section is now well-organized and seamlessly connected, with improved English that enhances the article's readability and overall comprehension. The most commendable aspect of the revision is the strengthened Conclusion as commented, which now includes more thorough and comprehensive implications, suggestions for future studies, and a clear direction for the way forward.

7. PLOS authors have the option to publish the peer review history of their article (what does this mean? ). If published, this will include your full peer review and any attached files.

**Do you want your identity to be public for this peer review?** For information about this choice, including consent withdrawal, please see our Privacy Policy .

Reviewer #2: No

---

## [Editor Report · Acceptance letter]

PONE-D-24-32451R1

PLOS ONE

Dear Dr. Nor,

I'm pleased to inform you that your manuscript has been deemed suitable for publication in PLOS ONE. Congratulations! Your manuscript is now being handed over to our production team.

Kind regards,

on behalf of

Dr. Xiaoxuan Kao

Guest Editor

PLOS ONE